# The complexity of NISQ

Sitan Chen[1,6] ✉, Jordan Cotler ●[2] ✉, Hsin-Yuan Huang ●[3,4] ✉ & Jerry Li[5] ✉

The recent proliferation of NISQ devices has made it imperative to understand their power. In this work, we define and study the complexity class NISQ, which encapsulates problems that can be efficiently solved by a classical computer with access to noisy quantum circuits. We establish super-polynomial separations in the complexity among classical computation, NISQ, and fault-tolerant quantum computation to solve some problems based on modifications of Simon's problems. We then consider the power of NISQ for three well-studied problems. For unstructured search, we prove that NISQ cannot achieve a Grover-like quadratic speedup over classical computers. For the Bernstein-Vazirani problem, we show that NISQ only needs a number of queries logarithmic in what is required for classical computers. Finally, for a quantum state learning problem, we prove that NISQ is exponentially weaker than classical computers with access to noiseless constant-depth quantum circuits.

Fault-tolerant quantum computing promises to offer speed-ups to various computational problems, including simulating quantum systems[1–5], factoring large numbers[6,7], performing optimization[8–11], and solving linear systems of equations[12]. While a fault-tolerant quantum computer has not yet been built, recent technological advancements have resulted in the development of new quantum devices that can outperform the best classical supercomputers for certain artificial computational tasks[13]. The opportunities offered by these devices and the challenges they engender have invigorated theorists and experimentalists alike, ushering in a new age of research into quantum computation and information often referred to as the NISQ (noisy intermediate-scale quantum) era[14].

Computation in the NISQ era is modeled by hybrid computation consisting of a classical computer and a noisy quantum device. Most currently available noisy quantum devices, such as those built from superconducting qubits[13,15–17], trapped ions[18–21], nuclear spins in silicon[22–24], or other solid-state systems[25–27], are restricted to preparing a noisy initial state, performing noisy quantum gates, and executing a noisy measurement on all qubits to generate a random bitstring. In most circumstances these noisy quantum devices are too weak to perform useful computation on their own. Generally, useful computation is facilitated by a classical computer that repeatedly runs the noisy quantum device with various gate sequences to obtain different classical output bit strings, and then performs classical post-processing on those strings. Algorithms within this computational model are referred to as hybrid quantum-classical algorithms[28,29] or NISQ algorithms[14,30].

The excitement around NISQ algorithms has led to a plethora of new near-term algorithms targeting different applications, including quantum chemistry[3,28,31–33], machine learning[17,34–39], combinatorial optimization[40–43], linear system solvers[44–46], and experimental data analysis[47–52]. However, to the best of our knowledge, no existing works have rigorously examined the class of all possible NISQ algorithms and studied their inherent computational power. As a result, many important and fundamental questions remain unanswered. In particular, how powerful are NISQ algorithms compared to classical algorithms? Are NISQ algorithms inherently weaker than fault-tolerant quantum algorithms?

In this work, we formalize and study these basic questions through the lens of computational complexity theory. To do so rigorously, we define a complexity class which we believe encapsulates what is possible on the vast majority of existing quantum devices. Our definition is intended to capture the following capabilities of noisy quantum devices, which we alluded to above:

1. Noisy quantum gates: The device can execute noisy two-qubit logic gates. Using quantum logic gates (as opposed to, say,

[1]Department of Electrical Engineering and Computer Science, University of California, Berkeley, Berkeley, CA, USA. [2]Society of Fellows, Harvard University, Cambridge, MA, USA. [3]Institute for Quantum Information and Matter, CAltech, Pasadena, CA, USA. [4]Department of Computing and Mathematical Sciences, CAltech, Pasadena, CA, USA. [5]Microsoft Research AI, Redmond, WA, USA. [6]Present address: John A. Paulson School of Engineering and Applied Sciences, Harvard University, Cambridge, USA. ✉e-mail: sitan@seas.harvard.edu; jcotler@fas.harvard.edu; hsinyuan@caltech.edu; jerrl@microsoft.com

more general non-unitary CPTP maps) is standard in existing quantum devices, and it is well-understood that in real-world settings, they will be subject to noise. For concreteness, we consider the standard model of local depolarizing noise per qubit. However, our results extend to more general noise models (see Remarks 2.4, 4.19 and 6.5) in the Supplementary Information.

2. Noisy state preparation at the start: The quantum devices have a fixed number of qubits and as such cannot bring in fresh qubits during the computation. This means that the device must prepare all qubits at the start. Notably, since we assume all quantum gates are subject to noise, this means all qubits will accrue entropy throughout the computation.

3. Noisy measurement at the end: The quantum devices are limited to perform noisy measurements only at the end of the computation, which means the measurement is performed on all qubits simultaneously. From a physical perspective, this constraint arises due to the difficulty of isolating subsets of qubits and measuring them without decohering the residual qubits.

Finally, we consider a classical computer that can repeatedly run the noisy quantum device and analyze the output from the noisy quantum device.

These constraints are chosen to encapsulate the gap between the physical limitations of what we can achieve with existing quantum computers and general quantum computation. We note these considerations preclude the implementation of all known general fault-tolerant quantum computation schemes[53–59], but that removing any one of these constraints would already allow for some form of non-trivial quantum fault tolerance[53,57,58]. The obstruction to fault tolerance can be understood intuitively. The noisy quantum gates cause all qubits to accrue entropy, which cannot be pumped out until the measurement at the end. Since too much entropy would destroy all useful quantum correlations, it is not possible for the noisy quantum devices under the above constraints to perform an arbitrarily long quantum computation. Note that in contrast, if we consider more benign noise which does not increase entropy, such as dephasing or amplitude-damping noise, then there are already schemes for achieving fault-tolerant quantum computation[57] within the above three constraints.

Motivated by the above considerations, in Section "Definition of NISQ" we formally define the NISQ complexity class to be the set of all problems that can be efficiently solved by a classical computer with access to a noisy quantum device that can (i) prepare a noisy poly($n$)-qubit all-zero state, (ii) execute noisy quantum gates, and (iii) perform a noisy measurement on all of the poly($n$) qubits. We subsequently show how the NISQ complexity class is situated relative to classical computers, and full-fledged quantum computers with quantum error correction.

## Results

In Section "Definition of NISQ" we give an overview of the definition of NISQ. Then, in Section "Super-polynomial separations", we give two modifications of Simon's problem which respectively yield a super-polynomial separation between BPP and NISQ, and an exponential separation between NISQ and BQP. In Section "NISQ in three well-studied problems", we study the NISQ complexity of three well-known problems: unstructured search, Bernstein-Vazirani problem, and shadow tomography. We defer all technical details to the Supplementary Information.

### Definition of NISQ

We begin with the definition of the NISQ complexity class, which is visually depicted in Fig. 1. The formal mathematical definition is given in Supplementary Note 2.A. In Supplementary Note 2.B, we recall the standard definition of oracle access in classical and quantum computation – when the oracle is classical, we extend the definition to NISQ by giving oracle access to both the classical computer and the noisy quantum computer.

**Definition 2.1.** (NISQ complexity class, informal) NISQ contains all problems that can be solved by a polynomial-time probabilistic classical algorithm with access to a noisy quantum device. To solve a problem of size $n$, the classical algorithm can access a noisy quantum device that can:

1. Prepare a noisy poly($n$)-qubit all-zero state;
2. Apply arbitrarily many layers of noisy two-qubit gates;
3. Perform noisy computational basis measurements on all the qubits simultaneously.

All quantum operations are subject to a constant amount of depolarizing noise per qubit.

The definition of a noisy quantum device given above forbids the implementation of all known fault-tolerant quantum computation schemes[53–56,59]. Hence, it is plausible that there are problems that could be solved efficiently by a fault-tolerant quantum algorithm but not by a NISQ algorithm, i.e. that NISQ ⊊ BQP.

The definition of NISQ immediately gives us that

$$\text{BPP} \subseteq \text{NISQ} \subseteq \text{BQP} . \tag{1}$$

The inclusion BPP ⊆ NISQ follows from the fact that a NISQ algorithm is a hybrid quantum-classical algorithm that can also run any classical computation. The latter inclusion NISQ ⊆ BQP holds because a quantum computer can simulate any noisy quantum device. However, it remains an open question whether BPP ⊊ NISQ and also if NISQ ⊊ BQP. The strict inclusion BPP ⊊ NISQ would imply that NISQ algorithms have a super-polynomial speedup over classical algorithms. The second strict inclusion NISQ ⊊ BQP would imply that NISQ

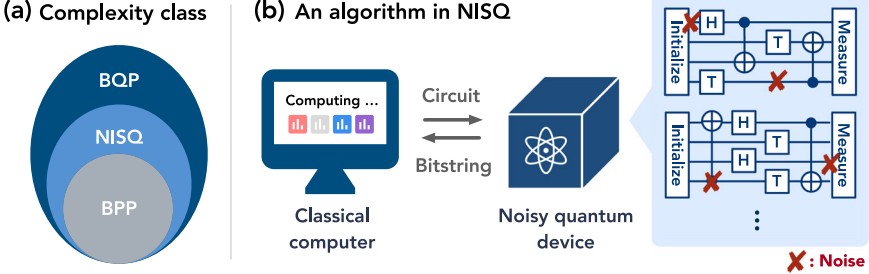

**Fig. 1 | Illustration of the NISQ complexity class. a** Complexity classes: NISQ contains problems that can be solved by classical computation (BPP), and some problems that can be solved by quantum computation (BQP). **b** NISQ algorithm: An algorithm in the complexity class NISQ is modeled by a hybrid quantum-classical algorithm, where a classical computer can specify the circuit to run on a noisy quantum device and the device would run a noisy version of the circuit and return a random classical bitstring obtained from noisy measurement.

algorithms are not as powerful as fault-tolerant quantum algorithms. Establishing either one of the strict inclusions would imply that BPP $\subsetneq$ BQP, which is a long-standing open problem in computational complexity theory. In this work, we follow the well-established approach in quantum complexity theory of proving oracle separations. That is, we consider various oracles, which are black boxes implementing certain functions or preparing some states, and show that NISQ algorithms accessing these oracles with noisy queries are strictly stronger or weaker than BPP or BQP algorithms that can access these boxes noiselessly. We consider widely studied oracles like the one for Grover search or for Bernstein-Vazirani problem, as well as variants of oracles like the one for Simon's problem.

### Super-polynomial separations

We begin with a set of results involving two modified versions of Simon's problem[60]. Simon's problem is one of the earliest examples of a computational problem demonstrating exponential quantum advantage in query complexity: given a function $f : \{0,1\}^n \mapsto \{0,1\}^n$, the goal is to decide if $f$ is 2-to-1 (with a promise that $f(x) = f(x \oplus s)$ for a secret string $s \in \{0,1\}^n$) or 1-to-1. In the language of computational complexity theory, it exhibits a classical oracle $O$ for which there is a relativized complexity separation between BPP and BQP, denoted by $\text{BPP}^O \subsetneq \text{BQP}^O$.

In this work, we give two modifications of Simon's problem that yield relativized separations among BPP, NISQ, and BQP. For NISQ, we consider noisy oracle access with local depolarizing noise occurring both before and after the oracle $O$. We first construct a modification of Simon's problem that requires at least a super-polynomial number of oracle queries for BPP and only a linear number for NISQ. We then give another modification of Simon's problem that requires at least an exponential number of queries for NISQ and only a linear number of queries for BQP. These two results can be summarized as follows:

**Theorem 2.2.** There is a classical oracle $O_1$ such that $\text{BPP}^{O_1} \subsetneq \text{NISQ}^{O_1}$.

**Theorem 2.3.** There is a classical oracle $O_2$ such that $\text{NISQ}^{O_2} \subsetneq \text{BQP}^{O_2}$.

Theorem 2.2 is established by constructing a robust version of the classical function $f : \{0,1\}^n \mapsto \{0,1\}^n$ in Simon's problem. We denote the robust version as $\widetilde{f} : \{0,1\}^{n'} \mapsto \{0,1\}^n$ with more input bits: $n' \gg n$. Each input $x \in \{0,1\}^n$ corresponds to a large set of inputs $A_x \subseteq \{0,1\}^{n'}$, such that every $z \in A_x$ produces the same output $\widetilde{f}(z) = f(x)$. The new function $\widetilde{f}$ is robust to noise, which allows a NISQ algorithm to achieve a super-polynomial speed-up.

The proof of Theorem 2.3 is essentially the opposite of that of Theorem 2.2. We construct a highly non-robust version $\widetilde{f}$ of the classical function $f$ in Simon's problem. Any noisy access to $\widetilde{f}$ provides exponentially little information, hence any NISQ algorithm would require exponentially more queries than a noiseless quantum algorithm. In fact, the separation we show is even stronger: not only is NISQ exponentially weaker than BQP relative to this oracle, but it is in fact even exponentially weaker than $\text{BPP}^{\text{QNC}^0}$, that is, classical computation assisted by noiseless bounded-depth quantum computation (see Supplementary Note 2.C for definitions).

Our findings give evidence that the computational power of NISQ lies somewhere strictly between BPP and BQP. The detailed proofs are given in Supplementary Note 4. Of course, separations relative to oracles come with the usual provisos inherent in relativized separations, like the fundamental complexity-theoretic barriers to deducing unconditional separations from relativized separations[61,62]. Another challenge is to instantiate the oracle in practice. While it is impossible to implement the relevant oracles perfectly on NISQ devices, if the oracle can be instantiated with cryptographic functions based on shallow circuits[63], then the techniques for proving Theorem 2.2 allow one to implement a robust version of the oracle in NISQ. Despite these

limitations, we view our results as promising evidence that NISQ may be truly intermediate between BPP and BQP.

### NISQ in three well-studied problems

Since the modified Simon's problems in Theorems 2.2 and 2.3 are tailored for proving super-polynomial separations, we would also like to study NISQ for more natural problems. We explore three well-known problems in quantum computing: unstructured search, the Bernstein-Vazirani problem, and shadow tomography.

For unstructured search, it is well-known that Grover's algorithm[64] can achieve a quadratic quantum speedup over any classical algorithm. To find a single marked element among $N$ elements, Grover's algorithm only requires $\mathcal{O}(\sqrt{N})$ queries, whereas any classical algorithm requires at least $\Omega(N)$ queries in the worst case. A natural open question is whether a NISQ algorithm can also achieve such a quadratic speedup. We resolve this open question in the negative by proving the following theorem. The proof is given in Supplementary Note 5.

**Theorem 2.4.** (Unstructured search) To find a single marked element among $N$ elements, any NISQ algorithm with access to poly($N$) qubits requires at least $\widetilde{\Omega}(N)$ queries.

The $\widetilde{\Omega}(\cdot)$ neglects logarithmic factors. We stress that the above theorem implies not only that noisy implementation of Grover's algorithm fails to achieve a quadratic speedup but, in fact, that any NISQ algorithm will fail to do so. The intuition behind the proof is that noisy quantum devices can only run for so long before noise overwhelms the system, so it would suffice to prove the lower bound for hybrid quantum-classical algorithms with access to a noiseless quantum device that can run any bounded-depth circuit. While a single run of any noiseless bounded-depth circuit cannot achieve Grover speedup[65], hybrid quantum-classical algorithms can perform many runs of noiseless bounded-depth circuits that depend adaptively on previous measurement outcomes. We prove that Grover speedup is impossible with hybrid quantum-classical algorithms using tools developed recently in the context of lower bounds for learning quantum states and processes using adaptive single-copy measurements[47–50,66–68].

For the second task, the Bernstein-Vazirani problem, we find that a large quantum advantage still remains for NISQ algorithms. Given an unknown $n$ bit string $s \in \{0,1\}^n$, the Bernstein-Vazirani problem asks how many queries to a function $f(x) = x \cdot s$ are required to learn $s$. Any classical algorithm requires at least $\Omega(n)$ queries to learn $s$. However, the Bernstein-Vazirani quantum algorithm can learn the unknown bit string $s$ with just 1 query. We show that the query complexity of this problem in NISQ remains much smaller than the classical query complexity:

**Theorem 2.5.** (Bernstein-Vazirani) There is a NISQ algorithm that solves the Bernstein-Vazirani problem over $n$ bits in at most $\mathcal{O}(\log n)$ queries.

Perhaps surprisingly, our analysis shows that a simple modification of the original Bernstein-Vazirani algorithm is already sufficiently noise-robust. This result provides optimism that there may be other natural problems for which quantum advantage can be obtained in the NISQ era. The detailed proof of Theorem 2.5 is presented in Supplementary Note 6.

Finally, we consider the problem of predicting many properties in an unknown quantum system, also known as shadow tomography[69–72]. In particular, we restrict to predicting absolute values of Pauli observables $\{I, X, Y, Z\}^{\otimes n}$: Given many copies of an unknown $n$-qubit state $\rho$, the goal is to learn $|\text{tr}(P\rho)|$ for all $P \in \{I, X, Y, Z\}^{\otimes n}$ up to a constant error by processing the quantum state copies. This task has received considerable attention in recent works[37,48,50,71] which have demonstrated, both theoretically and experimentally, that very simple BQP algorithms can solve this task using only $O(n)$ copies, while any classical

algorithm that can obtain classical data by performing measurements on individual copies of $\rho$ requires $\Omega(2^n)$ copies. Here we show that this large quantum advantage is damped by the presence of noise. Specifically, we show the following exponential separation between NISQ and BQP. The proof of the following theorem is given in Supplementary Note 7.

**Theorem 2.6.** (Shadow tomography) Any NISQ algorithm with noise rate $\lambda$ per qubit requires at least $\Omega((1-\lambda)^{-n})$ copies of $\rho$ to learn $|\text{tr}(P\rho)|$, for all $P \in \{I, X, Y, Z\}^{\otimes n}$ up to a constant error. In contrast, at most $\mathcal{O}(n)$ copies are needed for BQP.

Theorem 2.6 demonstrates that a fault-tolerant quantum algorithm can be exponentially more powerful than any NISQ algorithm in learning quantum systems. On the other hand, for small noise rate $\lambda$, the exponential scaling in the lower bound for NISQ algorithms has a base which is close to one. In[49] it was shown that NISQ algorithms can achieve $(1-\lambda)^{-\Theta(n)}$ even for more general noise models. This suggests that NISQ algorithms can still perform well for learning quantum systems with a few hundred qubits[50].

## Discussion

By abstracting hybrid quantum-classical computation in the NISQ era into a computational complexity class, our work offers a mathematical framework for reasoning about the potential for noisy quantum advantage. We used tools from quantum query complexity to characterize how NISQ lies between BPP and BQP. On the one hand, the fact that NISQ can be more powerful than BPP provides optimism for the NISQ era and the computational advances it may precipitate. On the other hard, NISQ being less powerful than BQP portends that we will have to wait until the advent of fault-tolerant devices to harness the richest features of quantum computation. Our results for the NISQ complexity of three well-known problems in quantum computing punctuate our outlook by demonstrating specific promises and pitfalls of computation in the NISQ era.

There are many future directions to pursue with the NISQ complexity class. A main open problem is to understand if we could show a separation between BPP, NISQ, and BQP under a standard complexity-theoretic assumption. It would be desirable to have an exponential oracle separation between BPP and NISQ, as opposed to one that is merely super-polynomial. Moreover, one could ask if a similar oracle separation exists between BPP and NISQ with the additional restriction that our quantum gates are spatially local, e.g., the gates are restricted to a two-dimensional geometry. Perhaps under this additional restriction of geometric locality on NISQ, it is possible to better understand the computational complexity of NISQ without relying on oracle separations. Additionally, it would be valuable to analyze the NISQ complexity of other natural quantum algorithms, beyond the ones we studied. Natural targets include the original Simon's problem (as opposed to our variations thereof, see Supplementary Note 10 for one approach in this direction), Forrelation[73], Shor's algorithm[6,7], linear system solving[12], the recent random oracle result of[74], and topological data analysis[75], among many others.

Taking a broader view, our work suggests a roadmap for investigating future quantum devices which may go beyond the NISQ era, but fall short of fault-tolerance. The approach is to formulate a computational complexity class which encapsulates the salient features of whichever quantum devices are contemporary, and then to study that complexity class to make statements about the capabilities of those devices with great generality. In such a future, there should be complexity classes beyond NISQ, but still intermediate to BQP. Whichever generalizations of NISQ prove fruitful in the future, they will have much to tell us about what computation is possible in that future world.

## Reporting summary

Further information on research design is available in the Nature Portfolio Reporting Summary.

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

## Acknowledgements
We would like to thank Isaac Chuang for valuable conversations on complexity classes for learning theory, John Preskill for inspiring discussions on NISQ and its complexity class formulation, and John Wright for bringing[76] and related works to our attention. S.C. is supported by NSF Award 2103300. J.C. is supported by a Junior Fellowship from the Harvard Society of Fellows, as well as in part by the Department of Energy under grant DE-SC0007870. H.H. is supported by a Google Ph.D. fellowship.

## Author contributions
S.C., J.C., H.H., and J.L. contributed equally to this work. All authors developed the theoretical aspects of this work and wrote the manuscript together.

## Competing interests
The authors declare no competing interests.
