## [Peer Review File · Nature Communications]

The Complexity of NISQREVIEWER COMMENTS

Reviewer #1 (Remarks to the Author):

The main contribution of this manuscript is the introduction of a new computational complexity class, called NISQ . The definition of this new class is intended to encapsulate decision problems that can be solved efficiently by classical computers that have access to near-term quantum devices. When designing this complexity class, the authors take into account some of the capabilities and limitations of noisy quantum devices, which comprise noisy gates, noisy inputs with a lack of fresh qubits, and noisy measurements.

This manuscript contains a number of novel results about the new complexity class NISQ and its relationship to other classical and quantum complexity classes. First, it proves oracle separations between BPP and NISQ, as well as between NISQ and BQP. Second, it proves that there is no quadratic speedup for unstructured search in NISQ. Third, it proves an exponential separation between NISQ and BQP for the Bernstein-Vazirani problem. Fourth, it proves an exponential separation between NISQ and BQP algorithms for the task of shadow tomography.

One chief concern I have about this manuscript is whether the complexity class NISQ actually encapsulates problems that we can expect to solve in the noisy intermediate-scale quantum (NISQ) era. While the class NISQ is motivated by some limitations of current quantum devices (namely, the presence of noise, lack of fresh qubits during the computation, and lack of adaptivity), it is not clear to me that it accounts for realistic models of noise in actual NISQ devices. For example, the simple model used here of local depolarizing noise acting independently on each qubit fails to describe more realistic noise models that might not be Markovian, time-independent, or gate-independent. In this regard, while the complexity class NISQ is an interesting class that can be analyzed theoretically, it might not be as physically relevant as it is claimed to be. In particular, it is not clear to me that it "encapsulates what is possible on the vast majority of existing quantum devices".

Second, while one could find scenarios where local depolarizing noise is a good approximation to realistic noise models, it seems physically unrealistic to describe noise as a tensor product of local depolarizing channels all with the same noise parameter λ . One might instead expect that bare wires, single-qubit gates, and two-qubit gates experience different levels of noise. Would it not be more natural, for the definition of the class NISQ (Definition 2.7), to allow the different local noise channels to take different noise parameters, and instead take λ to be the maximum over all these different noise parameters?

My previous point segues into my next point about Remarks 2.4, 4.19 and 6.5. While Definition 2.3, 2.5, 2.6, and 2.7 work with the single-qubit depolarizing channel, the authors in fact prove upper bounds with respect to a more general noise model, which is arguably more physically realistic. In this case, why not rework definitions 2.3, 2.5, 2.6, and 2.7 so that they instead take into account the more physically relevant noise model, instead of relegating its discussion to the three separate remarks? One could then define, say, the class NISQ in Definition 2.7 as a special case of this more general class. Since there is some arbitrariness involved in how one might define a complexity class to describe near-term hybrid quantum-classical algorithms, it might be preferable to have separate definitions for these different variants of NISQ .

Next, in definition 2.6 of the complexity class NISQ , the noise level λ is allowed to take any constant positive value, even very small ones. For very small values of λ , it would appear that the NISQ algorithm involved would behave almost like a BQP algorithm. Perhaps the authors could investigate whether it is the case that for small and perhaps physically relevant problem instances, a NISQ algorithm is almost no different from a BQP algorithm.

The class NISQ defines an interesting intermediate model of quantum computation, whose power lies in between that of classical computing and quantum computing. Relative to oracles, one can prove that NISQ is truly intermediate between BPP and BQP. One should note, however, the caveat that while the above separations between BPP, NISQ and BQP have been shown to hold in a relativized world, they do not necessarily

hold in our unrelativized world. Phrased in this way, it means that the oracle separations may actually tell us little about whether these classes are truly distinct in the real world, which is presumably a question that would be of greater interest especially in this NISQ era. While proving unconditional separations is difficult and out of the scope of this study, perhaps it would be helpful if the authors clearly delineated the limitations of using "tools from quantum query complexity to characterize how [unrelativized versions of] NISQ lies between BPP and BQP".

Overall, the manuscript is detailed and well written. I appreciate the effort that the authors go into explaining the motivation behind this work, and in carefully spelling out the concepts involved in the proofs of the main theorems. I believe that the results contained herein will be of interest to the quantum complexity community and is a good starting point for further research into characterizing the capabilities of near-term devices using tools from computational complexity theory. That said, I believe that the aforementioned points will need to be addressed first before one could consider it for publication in Nature Communications.

Some minor comments:

1. Since the entire of Section 7 is on shadow tomography, I think it might be helpful to the reader to include a section introducing shadow tomography and what is known about their performance bounds in realistic NISQ settings, in the Related Work section (section 1) of the Supplementary information.
2. On page 15, in definition 2.2, n is said to satisfy " $n > 0$ "; in definition 2.3, n is said to satisfy " $n \in \mathbb{N}$ "; and in definition 2.5, n is said to be an "integer". Since it appears that the authors are referring to the same domain for n in each of these cases, why not use more consistent terminology?
3. In Definition 2.5, "...a sequence of depth-1 n -qubit unitary" should instead be "...a sequence of depth-1 n -qubit unitaries".

4. In the first paragraph of section 2.C, "Sections 2" should be "Section 2".
5. On page 26, "similarly to in the proof..." should probably be "similarly to the proof..."
6. Equation S19 is missing a quantity that is to be maximized over.

Reviewer #2 (Remarks to the Author):

This submission defines a new complexity class inspired by near-term quantum devices, which they call "NISQ". The basic idea is to consider noisy quantum computations without the ability to inject fresh ancillas in the system, thus removing the ability of the quantum device to achieve fault tolerance and remove entropy from the system. Intuitively such a system accrues entropy and the computation becomes meaningless beyond a constant depth.

The authors show several new oracle results, which are variants of prior oracle separations between BPP and BQP. Most interestingly, they show that there are variants of Simon's problem which are insensitive to noise (and hence exhibit a NISQ speedup over BPP), created with error correcting codes. On the other hand, they define another variant of Simon's problems which is highly sensitive to noise and for which noise destroys the quantum speedup (and hence demonstrates a BQP speedup over NISQ). The authors also show several related results: namely that Bernstein-Vazirani can still be performed in NISQ, but efficient shadow tomography and Grover search cannot exhibit NISQ speedups. I did not find the latter result surprising as it has been long known, due to work of Zalka, that constant-depth quantum search offers no speedup, and therefore wonder if this might be derivable from prior results combined with bounds on the entropy accumulation in NISQ algorithms.

While these results are of reasonable interest, in my opinion the major limitation is that the authors assume that oracle evaluation is perfect/noiseless, while only the local gates applied in the computation are noisy. It is very important that the authors discuss this upfront (together with their motivations for the choice), as it's currently quite difficult to

infer this from the article. In my view this limits the claims that these problems can be used to exhibit an implementable NISQ speedup, as instantiating these oracles would themselves yield noise scaling with the depth, which may cause their algorithms to fail. This therefore might limit the applicability of their NISQ algorithms. On the other hand, this strengthens the authors' no-go results for NISQ speedups for other problems. Of course many works have also considered the opposite regime (noisy function evaluation but perfect gates, with various no-go results), and this paper can be seen as complementary to those prior results.

Despite this issue, I found the NISQ algorithmic speedup results (e.g. noise-resilient variants of Simon's algorithm) interesting. One's prior might be that there's nothing of interest in NISQ class, or that it's equivalent to studying constant depth quantum computation, so I found the work identifying interesting noise-robust problems interesting, even if in a slightly artificial setting.

Another issue I have is the assumption that noise models in near-term quantum experiments are necessarily entropy increasing – it would be nice if the authors had a more comprehensive, experimentally relevant, discussion on the viability of this assumption in different experimental architectures e.g., it's clear that if we model quantum noise purely as depolarizing noise then this is definitely the case, which is often a reasonable assumption, but what about, e.g., linear optical architectures where the primary source of noise – photon loss – can actually *decrease* entropy? Shouldn't we incorporate these architectures in a broad NISQ complexity class definition? I ask the authors to discuss this in the article of the paper (rather than the SI) in greater detail.

Reviewer #3 (Remarks to the Author):

This manuscript defines a formal model of NISQ computing that accommodates oracle problems. The model is based on treating the oracles as noiseless unitaries and the computation as built from noiseless depth-1 quantum circuits. In between any pair of noiseless operations is a layer of depolarizing channels, one on each qubit.

This is a reasonable abstract framework, which enables the authors to prove some

complexity-theoretic results about the power of NISQ devices. In particular, they find that in the presence of this noise model the Grover speedup goes away while the Bernstein-Vazirani speedup essentially remains.

The formal model is of course not fully realistic. In particular, real oracles will be implemented as quantum circuits themselves, e.g. reversible circuits for a typical Grover problem. Thus, the amount of noise incurred for each invocation of the oracle will depend on the number of gates needed to implement the oracle. It would not really look like a single layer of depolarizing channels applied after the oracle query.

Ultimately, I find that the formal model proposed by the authors, while not capturing full physical nuances, strikes a reasonable balance between physical realism and mathematical simplicity, which enables them to derive interesting, nontrivial, and informative results about a topic that is of great interest and very timely.

The paper is also written clearly and I do not detect any technical errors. I would recommend that it be accepted for publication without further revision.

Reviewer #1 (Remarks to the Author):

The main contribution of this manuscript is the introduction of a new computational complexity class, called NISQ. The definition of this new class is intended to encapsulate decision problems that can be solved efficiently by classical computers that have access to near-term quantum devices. When designing this complexity class, the authors take into account some of the capabilities and limitations of noisy quantum devices, which comprise noisy gates, noisy inputs with a lack of fresh qubits, and noisy measurements.

This manuscript contains a number of novel results about the new complexity class NISQ and its relationship to other classical and quantum complexity classes. First, it proves oracle separations between BPP and NISQ, as well as between NISQ and BQP. Second, it proves that there is no quadratic speedup for unstructured search in NISQ. Third, it proves an exponential separation between NISQ and BQP for the Bernstein-Vazirani problem. Fourth, it proves an exponential separation between NISQ and BQP algorithms for the task of shadow tomography.

Author response: We thank the reviewer for remarking on our introduction of a new complexity class which takes into account capabilities and limitations of noisy quantum devices, the novelty of our results, and the many separations we prove in our work.

One chief concern I have about this manuscript is whether the complexity class NISQ actually encapsulates problems that we can expect to solve in the noisy intermediate-scale quantum (NISQ) era. While the class NISQ is motivated by some limitations of current quantum devices (namely, the presence of noise, lack of fresh qubits during the computation, and lack of adaptivity), it is not clear to me that it accounts for realistic models of noise in

actual NISQ devices. For example, the simple model used here of local depolarizing noise acting independently on each qubit fails to describe more realistic noise models that might not be Markovian, time-independent, or gate-independent. In this regard, while the complexity class NISQ is an interesting class that can be analyzed theoretically, it might not be as physically relevant as it is claimed to be. In particular, it is not clear to me that it “encapsulates what is possible on the vast majority of existing quantum devices.”

Author response:

We have designed the NISQ complexity class to provide a realistic model of contemporary quantum devices (as of the writing of the paper) and have vetted the definition with originators of the NISQ concept including John Preskill.

At a technical level, we would like to re-emphasize two points. First, from the standpoint of *lower bounds*, the simplicity of our model is actually a strength of our work as we are able to show strong impossibility results even for a noise model as simple and canonical as local depolarizing noise. Naturally, such results will continue to hold under the more sophisticated noise models the reviewer mentions. Second, from the standpoint of *upper bounds*, as the referee notes later in the review, our results apply even to the more physically realistic setting where we simply assume every qubit is corrupted with some probability by an adversary. This is also precisely the noise model considered in the celebrated threshold theorem for fault-tolerant quantum computation.

Remark A.4 in the submission, which is mentioned later in this review and which we have reproduced below, succinctly summarizes the above points:

“Remark A.4. We work with the single-qubit depolarizing channel as it is the most standard model for local noise. One could also consider stronger noise models, e.g. every qubit is randomly corrupted with probability λ by an adversary rather than randomly decohered. Tautologically, the lower bounds we prove in this work will translate to such stronger models. We also prove our upper bounds, namely Theorem 2.2 and 2.5, under this stronger model (see Remarks C.19 and E.5).”

Ultimately, we centered our definition for NISQ around local depolarizing noise primarily for the purposes of exposition, in order to have a conceptually clean “base definition” to work with, but the strength of our results does not suffer from this choice. Our definition is also sufficiently modular that researchers interested in how the complexity landscape changes under, e.g., non-Markovian noise, can easily mix-and-match noise models of their choice with our base definition.

Second, while one could find scenarios where local depolarizing noise is a good approximation to realistic noise models, it seems physically unrealistic to describe noise as a tensor product of local depolarizing channels all with the same noise parameter λ . One might instead expect that bare wires, single-qubit gates, and two-qubit gates experience different levels of noise. Would it not be more natural, for the definition of the class NISQ (Definition 2.7), to allow the different local noise channels to take different noise parameters, and instead take λ to be the maximum over all these different noise parameters?

Author response: This is not a problem, essentially for the same reasons as outlined above. Our lower and upper bounds straightforwardly extend to the setting where we take

different amounts of depolarizing noise λ per qubit. For the upper bounds, this setting is just a special case of the stronger adversarial noise model under which our upper bounds already hold. For the lower bounds, note that one would need to assume a *minimum* over λ 's for it to make sense to prove an impossibility result. If the minimum is λ_{\min} , then our impossibility results for when the noise rate is uniformly λ_{\min} immediately imply impossibility results for when the noise rate is at least this much.

My previous point segues into my next point about Remarks 2.4, 4.19 and 6.5. While Definition 2.3, 2.5, 2.6, and 2.7 work with the single-qubit depolarizing channel, the authors in fact prove upper bounds with respect to a more general noise model, which is arguably more physically realistic. In this case, why not rework definitions 2.3, 2.5, 2.6, and 2.7 so that they instead take into account the more physically relevant noise model, instead of relegating its discussion to the three separate remarks? One could then define, say, the class NISQ in Definition 2.7 as a special case of this more general class. Since there is some arbitrariness involved in how one might define a complexity class to describe near-term hybrid quantum-classical algorithms, it might be preferable to have separate definitions for these different variants of NISQ.

Author response: We thank the reviewer for pointing this out. We had actually originally taken this approach, writing the manuscript with definitions reflecting the most general noise model, but found that it hindered its readability and accessibility with colleagues. Our colleagues suggested that we instead use the more limited definition so that the proofs and definitions were clearer and could be more readily grasped, and then note the generalizations along the way. As such, we opted for this stylistic choice, but we appreciate the idea of alternative approaches to presentation.

Next, in definition 2.6 of the complexity class NISQ, the noise level λ is allowed to take any constant positive value, even very small ones. For very small values of λ , it would appear that the NISQ algorithm involved would behave almost like a BQP algorithm. Perhaps the authors could investigate whether it is the case that for small and perhaps physically relevant problem instances, a NISQ algorithm is almost no different from a BQP algorithm.

Author response: Note that in the main text, all of our lower bounds are actually stated and proven not just for constant λ but for *any* λ , in the sense that we actually make explicit the quantitative dependence on λ without assuming any asymptotic lower bounds on its magnitude. For instance, for Grover search, our lower bound scales as $d\lambda$, so provided that $\lambda \gg 1/\sqrt{d}$, one cannot get the full Grover speedup in NISQ. The intuition roughly speaking is that by Lemma D.15, the amount of information at the T -th layer decays as $(1 - \lambda)^T \cdot n$, where n is the number of qubits. This means that for any noise rate $\lambda \gg 1/T$, the amount of information at that layer is still exponentially small.

As another example, our oracle separation between NISQ and BQP yields an $\exp(\lambda n)$ lower bound for solving our modified Simon's problem in NISQ, so even when $\lambda \asymp 1/n^{0.9}$, there is still an exponential lower bound.

That said, certainly if λ is sufficiently small, e.g. $1/N$ where N is the number of qubits multiplied by the width of the quantum circuit being noisily implemented, then with constant probability, no noise gets applied and the reviewer's intuition is indeed correct that

the computation amounts to noiseless quantum computation.

The class NISQ defines an interesting intermediate model of quantum computation, whose power lies in between that of classical computing and quantum computing. Relative to oracles, one can prove that NISQ is truly intermediate between BPP and BQP. One should note, however, the caveat that while the above separations between BPP, NISQ and BQP have been shown to hold in a relativized world, they do not necessarily hold in our unrelativized world. Phrased in this way, it means that the oracle separations may actually tell us little about whether these classes are truly distinct in the real world, which is presumably a question that would be of greater interest especially in this NISQ era. While proving unconditional separations is difficult and out of the scope of this study, perhaps it would be helpful if the authors clearly delineated the limitations of using “tools from quantum query complexity to characterize how [unrelativized versions of] NISQ lies between BPP and BQP”.

Author response: The reviewer notes that the complexity class separations we prove are only relative to oracles. After the discussion at the end of Section 2.1 in which we point out that obtaining unconditional separations between NISQ and either BPP or BQP would imply $BPP \neq BQP$ which is too much to hope for, we included some prose stating that we instead follow the *well-established* approach in quantum complexity of proving relativized separations instead. We also added prose noting the usual caveats with such separations.

We note, however, that such caveats do not apply to all of our results. In particular, our lower bound for shadow tomography is relative to a *quantum state oracle*. In the context of using near-term devices to learn about the physical world, it is natural and indeed necessary to assume access to such an oracle as a way of interfacing the device with the physical process that is being studied or experimented upon.

Overall, the manuscript is detailed and well-written. I appreciate the effort that the authors go into explaining the motivation behind this work and in carefully spelling out the concepts involved in the proofs of the main theorems. I believe that the results contained herein will be of interest to the quantum complexity community and is a good starting point for further research into characterizing the capabilities of near-term devices using tools from computational complexity theory. That said, I believe that the aforementioned points will need to be addressed first before one could consider it for publication in Nature Communications.

Author response: We are encouraged that the referee found our manuscript to be detailed and well-written. We thank them for recognizing our efforts in explaining motivations and key concepts in the proofs and theorems so that our work is more broadly accessible. We feel that, having addressed the referees’ aforementioned concerns both via this rebuttal and via the modifications made to the manuscript, our paper is suitable for publication in *Nature Communications*.

Some minor comments:

1. *Since the entire of Section 7 is on shadow tomography, I think it might be helpful to the reader to include a section introducing shadow tomography and what is known about their performance bounds in realistic NISQ settings, in the Related Work section (section 1) of the Supplementary information.*

2. *On page 15, in definition 2.2, n is said to satisfy “ $n > 0$ ”; in definition 2.3, n is said to*

satisfy $n \in \mathbb{N}$; and in definition 2.5, n is said to be an "integer". Since it appears that the authors are referring to the same domain for n in each of these cases, why not use more consistent terminology?

3. In Definition 2.5, "...a sequence of depth-1 n -qubit unitary" should instead be "...a sequence of depth-1 n -qubit unitaries".
4. In the first paragraph of section 2.C, "Sections 2" should be "Section 2".
5. On page 26, "similarly to in the proof..." should probably be "similarly to the proof..."
6. Equation S19 is missing a quantity that is to be maximized over.

Author response:

We thank the referee for pointing these out and for their careful reading of our manuscript. We have implemented all of these changes. We have also added an introduction to shadow tomography and discussed what is known about performance bounds in NISQ-like settings, e.g. when the device only has a limited amount of quantum memory. Given the length and the fact that this discussion is only pertinent to Section 7 of our manuscript, we chose to add this to the beginning of Section 7.

Reviewer #2 (Remarks to the Author):

This submission defines a new complexity class inspired by near-term quantum devices, which they call "NISQ". The basic idea is to consider noisy quantum computations without the ability to inject fresh ancillas in the system, thus removing the ability of the quantum device to achieve fault tolerance and remove entropy from the system. Intuitively such a system accrues entropy and the computation becomes meaningless beyond a constant depth.

The authors show several new oracle results, which are variants of prior oracle separations between BPP and BQP. Most interestingly, they show that there are variants of Simon's problem which are insensitive to noise (and hence exhibit a NISQ speedup over BPP), created with error correcting codes. On the other hand, they define another variant of Simon's problems which is highly sensitive to noise and for which noise destroys the quantum speedup (and hence demonstrates a BQP speedup over NISQ). The authors also show several related results: namely that Bernstein-Vazirani can still be performed in NISQ, but efficient shadow tomography and Grover search cannot exhibit NISQ speedups. I did not find the latter result surprising as it has been long known, due to work of Zalka, that constant-depth quantum search offers no speedup, and therefore wonder if this might be derivable from prior results combined with bounds on the entropy accumulation in NISQ algorithms.

Author response: We thank the reviewer for summarizing the results of our work. In light of the reviewer's remark about the Grover result, we would like to clarify what we believe to be an important misunderstanding.

The work of Zalka shows that a running a *single*, noiseless bounded-depth quantum circuit cannot solve the unstructured search problem as the output state would not be sufficiently distinct. The entropy accumulation result shows that a single run of a noisy quantum circuit will not be much more powerful than a noiseless bounded-depth quantum circuit.

Combining the Zalka result and the entropy accumulation shows that a single run of a noisy quantum circuit is not enough to achieve Grover speedup. We believe this is what the reviewer has in mind, and this result is indeed straightforward to prove. However, *NISQ algorithms can be much more powerful than a single run of a noisy quantum circuit. Indeed, the main content of our impossibility result is precisely that even with this significant added power, NISQ algorithms cannot realize Grover speedups.*

To be more precise, recall that in NISQ algorithms, one can run many rounds of noisy quantum circuits. The quantum circuit in each round can depend adaptively on the measurement outcomes of previous rounds of quantum circuits. This hybrid quantum-classical nature could provide stronger computational power than a single round of bounded-depth quantum computation. For example, various positive results on NISQ algorithms shown in our work use additional computational power by running multiple rounds of noisy quantum circuits. Hence, one needs to show that having these more powerful hybrid quantum-classical algorithms with many rounds of noisy quantum computations is still not enough to achieve Grover speedups. To do so, we utilize a new set of ideas developed recently in the context of lower bounds for learning quantum states and processes using unentangled measurements. The connection between the lower bounds of hybrid quantum-classical computation and of learning quantum states is a key finding of this work, which we believe to be nontrivial and interesting.

We have updated the discussion after Theorem 2.4 in the manuscript to summarize the above discussion.

While these results are of reasonable interest, in my opinion the major limitation is that the authors assume that oracle evaluation is perfect/noiseless, while only the local gates applied in the computation are noisy. It is very important that the authors discuss this upfront (together with their motivations for the choice), as it's currently quite difficult to infer this from the article. In my view this limits the claims that these problems can be used to exhibit an implementable NISQ speedup, as instantiating these oracles would themselves yield noise scaling with the depth, which may cause their algorithms to fail. This therefore might limit the applicability of their NISQ algorithms. On the other hand, this strengthens the authors' no-go results for NISQ speedups for other problems. Of course many works have also considered the opposite regime (noisy function evaluation but perfect gates, with various no-go results), and this paper can be seen as complementary to those prior results.

Despite this issue, I found the NISQ algorithmic speedup results (e.g. noise-resilient variants of Simon's algorithm) interesting. One's prior might be that there's nothing of interest in NISQ class, or that it's equivalent to studying constant depth quantum computation, so I found the work identifying interesting noise-robust problems interesting, even if in a slightly artificial setting.

Author response: We appreciate the positive feedback and agree that the computational capabilities that could emerge in the NISQ complexity class are surprising. Here we address the reviewer's comments on noise and oracles.

We begin by clarifying a point which may not have been clear in the original manuscript. In the proofs, we consider noise to occur both before and after the oracle. Hence the oracles in

NISQ algorithms are still noisy (as observed by Reviewer 3). For example, in shadow tomography, it is actually precisely the noise in the state oracle, rather than the noise in the computation that occurs in between state oracle queries, which is the cause for the exponential slowdown compared to BQP. We have revised our manuscript to emphasize early on that noise occurs both before and after the oracles in NISQ.

We also note that our results are also somewhat robust to noise *within* the oracle. Indeed, prior work on noisy/faulty oracles has considered noise inside oracles to be given by a global depolarizing channel. While this choice is also debatable, having global depolarizing noise inside the oracle is actually more benign than having local noise in the quantum circuit. Local noise in the quantum circuit implies that the quantum circuit must be of bounded depth by the entropy accumulation argument. The bounded depth implies that additional global depolarizing noise inside the oracle will not incur too much noise. Hence, all of our results would still go through if there is global depolarizing noise inside the oracle. Additionally, some real-world instantiations of oracles (based on cryptographic hardness) can be implemented using only bounded depth. In this case, the oracles would also not incur too much noise. However, due to the various geometric constraints in current noisy quantum devices, how to run these NISQ algorithms in practice is still an open question.

*Another issue I have is the assumption that noise models in near-term quantum experiments are necessarily entropy increasing – it would be nice if the authors had a more comprehensive, experimentally relevant, discussion on the viability of this assumption in different experimental architectures e.g., it’s clear that if we model quantum noise purely as depolarizing noise then this is definitely the case, which is often a reasonable assumption, but what about, e.g., linear optical architectures where the primary source of noise – photon loss – can actually *decrease* entropy? Shouldn’t we incorporate these architectures in a broad NISQ complexity class definition? I ask the authors to discuss this in the article of the paper (rather than the SI) in greater detail.*

Author response: Thank you for raising this question. If we consider entropy-decreasing noise, such as dephasing or damping noise, there exist schemes for implementing fault-tolerant quantum computation using the noisy quantum device. Hence, if we replace our current definition of NISQ, which is based on entropy-increasing noise, with one that is based on entropy-decreasing noise, then we would immediately have $\text{NISQ} = \text{BQP}$. We have added a brief discussion in the article of the paper.

Reviewer #3 (Remarks to the Author):

This manuscript defines a formal model of NISQ computing that accommodates oracle problems. The model is based on treating the oracles as noiseless unitaries and the computation as built from noiseless depth-1 quantum circuits. In between any pair of noiseless operations is a layer of depolarizing channels, one on each qubit.

This is a reasonable abstract framework, which enables the authors to prove some complexity-theoretic results about the power of NISQ devices. In particular, they find that in the presence of this noise model the Grover speedup goes away while the Bernstein-Vazirani

speedup essentially remains.

The formal model is of course not fully realistic. In particular, real oracles will be implemented as quantum circuits themselves, e.g. reversible circuits for a typical Grover problem. Thus, the amount of noise incurred for each invocation of the oracle will depend on the number of gates needed to implement the oracle. It would not really look like a single layer of depolarizing channels applied after the oracle query.

Ultimately, I find that the formal model proposed by the authors, while not capturing full physical nuances, strikes a reasonable balance between physical realism and mathematical simplicity, which enables them to derive interesting, nontrivial, and informative results about a topic that is of great interest and very timely.

The paper is also written clearly and I do not detect any technical errors. I would recommend that it be accepted for publication without further revision.

Author response: We thank the reviewer for the positive feedback on our manuscript.

REVIEWERS' COMMENTS

Reviewer #1 (Remarks to the Author):

In this new version of the manuscript, the authors have implemented a number of changes, including:

- a note about more benign noise models which do not increase entropy, and how there already are schemes for these models that achieve fault-tolerant quantum computation, on page 2.
- prose noting caveats associated with oracle separations, in response to the referee's comment that separations in a relativized world do not necessarily hold in the unrelativized setting, on page 3.
- a note clarifying that for the noisy oracle, local depolarizing noise occurs both before and after the oracle, on page 4.
- some intuition behind the proof of Theorem 2.4, on page 5.

In addition, they have corrected a number of typos present in the previous version of the manuscript.

These changes are positive and they have improved the clarity and readability of the manuscript.

The authors have expressed their disagreement with the referee's suggestion of initially presenting a version of NISQ with the most general noise model, citing concerns that it might hinder readability and accessibility. While respecting the authors' prerogative to make these stylistic choices, I wonder whether the absence of more distinct and clearer definitions for the various NISQ variants might represent a missed opportunity. Having clearer definitions could enhance the manuscript's readability, as readers would not need to search for mentions of these variations in scattered remarks (Remarks 2.4, 4.19, and 6.5) throughout the text.

The authors write in their response letter: "That said, certainly if λ is sufficiently small, e.g. $1/N$ where N is the number of qubits multiplied by the width of the quantum circuit being

noisily implemented, then with constant probability, no noise gets applied and the reviewer's intuition is indeed correct that the computation amounts to noiseless quantum computation." While I found this discussion to be insightful, it does not appear to have been included in the revised version of the manuscript. I'd recommend incorporating this point into the manuscript for the readers' benefit.

Page 2's "such as dephasing or damping noise": I'm not sure what is meant by "damping noise". Are the authors referring to amplitude-damping noise?

On page 4, in the line "then the techniques for proving Theorem 2.2 allows one," "allows" should be replaced with "allow."

Reviewer #2 (Remarks to the Author):

I have reread the manuscript and the authors have done a satisfactory job at addressing the outstanding comments. I believe the current manuscript is suitable for publication.

Reviewer #1 (Remarks to the Author):

In this new version of the manuscript, the authors have implemented a number of changes, including: - a note about more benign noise models which do not increase entropy, and how there already are schemes for these models that achieve fault-tolerant quantum computation, on page 2. - prose noting caveats associated with oracle separations, in response to the referee's comment that separations in a relativized world do not necessarily hold in the unrelativized setting, on page 3. - a note clarifying that for the noisy oracle, local depolarizing noise occurs both before and after the oracle, on page 4. - some intuition behind the proof of Theorem 2.4, on page 5.

In addition, they have corrected a number of typos present in the previous version of the manuscript.

These changes are positive and they have improved the clarity and readability of the manuscript.

We thank the reviewer for the positive feedback on the changes we have implemented.

The authors have expressed their disagreement with the referee's suggestion of initially presenting a version of NISQ with the most general noise model, citing concerns that it might hinder readability and accessibility. While respecting the authors' prerogative to make these stylistic choices, I wonder whether the absence of more distinct and clearer definitions for the various NISQ variants might represent a missed opportunity. Having clearer definitions could enhance the manuscript's readability, as readers would not need to search for mentions of these variations in scattered remarks (Remarks 2.4, 4.19, and 6.5) throughout the text.

We agree that there are various tradeoffs in how we have chosen to present our NISQ definition. As our Remark 2.4 immediately follows our definition of noisy quantum circuits

in the section in which we first define the NISQ complexity class, we believe we make sufficiently clear the possibility of variations to our NISQ definition, but there are certainly alternative ways of presenting these variants that have complementary advantages.

The authors write in their response letter: “That said, certainly if λ is sufficiently small, e.g. $1/N$ where N is the number of qubits multiplied by the width of the quantum circuit being noisily implemented, then with constant probability, no noise gets applied and the reviewer’s intuition is indeed correct that the computation amounts to noiseless quantum computation.” While I found this discussion to be insightful, it does not appear to have been included in the revised version of the manuscript. I’d recommend incorporating this point into the manuscript for the readers’ benefit.

We have added a remark to this effect right after the definition of NISQ.

Page 2’s ”such as dephasing or damping noise”: I’m not sure what is meant by ”damping noise”. Are the authors referring to amplitude-damping noise?

Yes, we have clarified in the revision that we meant amplitude-damping noise.

On page 4, in the line ”then the techniques for proving Theorem 2.2 allows one,” ”allows” should be replaced with ”allow.”

We have fixed this.

Reviewer #2 (Remarks to the Author):

I have reread the manuscript and the authors have done a satisfactory job at addressing the outstanding comments. I believe the current manuscript is suitable for publication.

We thank the reviewer for their time in engaging with our work and for their positive feedback.